# Targeting Myeloid-Derived Suppressor Cells in Ovarian Cancer

**DOI:** 10.3390/cells10020329

**Published:** 2021-02-05

**Authors:** Seiji Mabuchi, Tomoyuki Sasano, Naoko Komura

**Affiliations:** 1Department of Obstetrics and Gynecology, Nara Medical University, Nara 634-8522, Japan; 2Department of Obstetrics and Gynecology, Osaka Saiseikai Nakatsu Hospital, Osaka 530-0012, Japan; sasano106@yahoo.co.jp; 3Department of Obstetrics and Gynecology, Kaizuka City Hospital, Osaka 597-0015, Japan; naonaokomura@gmail.com

**Keywords:** MDSC, ovarian cancer, survival, therapeutic target, tumor microenvironment

## Abstract

Myeloid-derived suppressor cells (MDSCs) are a heterogeneous population of immature myeloid cells that exhibit immunosuppressive activity. They also directly stimulate tumor cell proliferation, metastasis, and angiogenesis. In ovarian cancer, there are increased numbers of circulating or tumor-infiltrating MDSCs, and increased frequencies of MDSCs are associated with a poor prognosis or an advanced clinical stage. Moreover, in murine models of ovarian cancer, MDSC depletion has shown significant growth-inhibitory effects and enhanced the therapeutic efficacy of existing anticancer therapies. In this review, we summarize the current knowledge on MDSC biology, clinical significance of MDSC, and potential MDSC-targeting strategies in ovarian cancer.

## 1. Introduction

Epithelial ovarian cancer is a leading cause of cancer-related death among women, accounting for 295,000 new cases and 185,000 deaths annually worldwide [1]. Due to its asymptomatic nature and lack of effective screening tests, most patients are diagnosed at advanced stages [2]. Although most advanced-stage ovarian cancers respond to the initial treatment, including primary debulking surgery and platinum-based chemotherapy, more than 70% of patients will ultimately relapse [3]. Therefore, it is imperative to overcome platinum resistance by identifying new therapeutic targets.

An increasing body of evidence suggests that the immunological microenvironment plays a significant role in the development/progression of cancer, and treatment options that activate the immune surveillance response have emerged as a promising cancer treatment strategy [4]. However, although immunotherapy has become a viable treatment for recurrent ovarian cancer, current immunotherapies face barriers that limit their clinical efficacy: only a limited number of patients have responded to checkpoint inhibitors, and antigen-specific active immunotherapy has demonstrated no survival benefit [5,6].

Various types of cancer can undergo intratumoral immunosuppression, including ovarian cancer [7,8]. This facilitates cancer cells to escape from destruction by the immune system, and could limit the therapeutic efficacy of current immunotherapies that target cytotoxic T lymphocyte-associated protein 4 (CTLA-4) or programmed death ligand 1 (PD-L1)/programmed death 1 (PD-1) [4].

In the early 20th century, it was noted that cancer progression was often accompanied by extramedullary hematopoiesis and resulting leukocytosis or neutrophilia [9]. These leukocytes or neutrophilia were further characterized by their suppressive activities and were called immature myeloid cells, myeloid suppressor cells, or natural suppressor cells. Eventually, they were named myeloid-derived suppressor cells (MDSCs) in 2007 [10]. Since then, an increasing amount of information regarding the biology and clinical significance of MDSCs in various pathological conditions has been reported. MDSCs are a heterogeneous population of immature myeloid cells (IMC) which can promote tumor growth by inducing tumor cell immunological anergy and tolerance; they block the proliferation and activity of both T cells and natural killer (NK) cells [11]. In addition, MDSCs can directly stimulate tumor cell proliferation, metastasis, and angiogenesis, all of which can lead to tumor progression and limit the potency of current therapeutic interventions [4,11]. As the acquired ability of cancer cells to escape from immune surveillance is a hallmark of ovarian cancer and increased MDSC levels have been demonstrated in ovarian cancer patients, MDSCs are now regarded as a promising therapeutic target and a predictive biomarker of treatment outcomes.

In this review, we summarize the current knowledge on MDSC biology and its role in ovarian cancer. We also discuss the utility of the number of MDSCs as a predictive marker and highlight how MDSCs can be targeted therapeutically in patients with ovarian cancer.

## 2. Definition of MDSC

MDSCs are a heterogenic population of IMC that differ in morphology and function from terminally differentiated mature myeloid cells (macrophages, dendritic cells (DC), or neutrophils). MDSC can be subdivided into two major subsets, monocytic MDSC (M-MDSC), which morphologically and phenotypically resemble monocytes, and polymorphonuclear (PMN) MDSC (also known as granulocytic MDSC), which are morphologically similar to neutrophils [10,11].

In mice, MDSCs were historically characterized by their expression of glutathione reductase (Gr-1) and CD11b myeloid lineage differentiation markers (CD11b^+^Gr-1^+^ cells). However, Gr-1 is not a single molecule, but a combination of the lymphocyte antigen (Ly)6C and Ly6G molecules; therefore, MDSCs can be more accurately identified based on these markers (M-MDSC are Ly6C^high^Ly6G^−^, and PMN-MDSC are Ly6C^low^Ly6G^+^) [10,11].

Although both mice and human MDSCs express CD11b, human MDSCs do not have the Gr-1 antigen (Ly6G/Ly6C) and are more complex. However, it is generally accepted that human MDSCs are positive for CD11b and CD33, and negative for the human leukocyte antigen-antigen D related (HLA-DR) and lineage markers (CD3, CD13, CD19, CD56). As in murine cells, human MDSCs can also be divided into two groups: PMN-MDSC, which generally express CD15 but lack CD14 (CD11b^+^CD33^+^CD14^−^CD15^+^ cells), and M-MDSC, which usually express CD14 but not CD15 (CD11b^+^CD33^+^CD14^+^CD15^−^ cells) [10,11]. To distinguish PMN-MDSC from neutrophils, lectin-type oxidized LDL receptor 1 (LOX-1) has been used; PMN-MDSC overexpress LOX-1 while neutrophils do not express LOX-1 [12]. In addition to abovementioned surface markers, the “gold standard” for defining MDSCs is based on their ability to inhibit T cells [11].

## 3. Functions of MDSCs

### 3.1. Immunosuppressive Functions of MDSCs

MDSCs (both PMN-MDSC and M-MDSC) can suppress both innate and adaptive immune responses. As shown in Figure 1, MDSCs mainly exert their suppressive effects by producing arginase-1 (Arg-1) which causes the removal of L-arginine, an essential amino acid for T cell differentiation, from the tumor microenvironment (TME). The depletion of L-arginine subsequently causes the downregulation of CD247 (the ζ-chain of the T cell receptor) expression in T cells. As CD247 is a subunit of the natural killer receptors NKp46, NKp30, and TcγIII in NK cells, the depletion of L-arginine leads to the inhibition of T cell and NK cell proliferation. MDSCs also produce reactive oxygen species (ROS) and nitric oxide (NO), which lead to the nitration of signaling molecules downstream of the FcγRIIIA, resulting in the inhibition of the activities of T cells and NK cells [10,11]. NO produced by MDSCs also nitrate signal transducers and activators of transcription (STAT)1, leading to the diminished interferon response in T cells and NK cells. Moreover, MDSCs induce regulatory T cell (Treg) expansion, which also acts to suppress the effector T cells [11]. Lastly, MDSCs have an increased expression of PD-L1, which leads to the downregulation of T cell function via engagement of cell surface PD-1 [13].

### 3.2. Nonimmune Activities of MDSC

In addition to suppressing immune responses within the TME, MDSCs also promote cancer progression by stimulating tumor angiogenesis and enhancing tumor cell invasion and metastasis (Figure 1). These processes are regulated by MDSC-derived mediators, including vascular endothelial growth factor (VEGF), basic fibroblast growth factor (bFGF), Bv8, and matrix metalloproteinase (MMP)-9, all of which are essential for tumor angiogenesis and cancer cell invasion [11,14]. Moreover, recent reports suggested that MDSCs stimulate the metastatic activity of cancer cells by facilitating epithelial-to-mesenchymal transition (EMT) or by creating “premetastatic niches” [15,16]. Importantly, the evidence has indicated that MDSCs induce “stemness,” which might be associated with resistance to existing anticancer treatments including chemotherapy or radiotherapy [17].

## 4. MDSC Generation and Recruitment

In healthy individuals, IMC develop from common myeloid progenitor cells that differentiate into mature nonsuppressive myeloid cells such as DCs, macrophages, or granulocytes. Under pathological conditions such as infection, inflammation, or cancer, their differentiation was directed away from mature nonsuppressive cells to suppressive cells (MDSCs) [10,11]. It has been reported that PMN-MDSC undergo expansion in most patients with solid tumors. However, in certain cancers, such as melanoma, multiple myeloma, and prostate cancer, the M-MDSC population is more prevalent [18].

### 4.1. MDSC Generation and Activation

In cancerous conditions, MDSC generation from myeloid precursors is stimulated by the various tumor-derived cytokines or growth factors. These factors include tumor necrosis factor α (TNF-α), S100A8/9 proteins, macrophage colony-stimulating factor (M-CSF), granulocyte colony-stimulating factor (G-CSF), granulocyte monocyte colony-stimulating factor (GM-CSF), VEGF, prostaglandin E2 (PGE2), and interleukins (IL-1β, IL-10, IL-18, and IL-6). These trigger the activation of signaling pathways, including the Janus kinase (JAK)/STAT3, phosphoinositide 3-kinase/AKT/mammalian target of rapamycin (PI3K/AKT/mTOR), Ras/mitogen-activated protein kinase (RAS/MAPK), nuclear factor-kappa B (NFκB), toll-like receptor signaling, and SMAD pathways [11]. Of these, the activation of STAT3 seems to be the most prominent because it contributes to the proliferation and differentiation of myeloid precursors into MDSCs and stimulates arginase production by binding directly to the Arg-1 promoter [19]. STAT3 also downregulates interferon-related factor (IRF)-8, and upregulates CCAAT-enhancer-binding protein (C/EBP)-β, both of which are crucial regulators of MDSC activity [20,21]. Moreover, it has been reported that tumor-derived PGE2 contributes to the DNA hypermethylation, which plays important roles in MDSC differentiation from myeloid precursors and in MDSC’s immunosuppressive activity [22].

### 4.2. Recruitment of MDSC into TME

An important factor directing the migration of MDSC is chemokines. Studies have shown that multiple chemokines in the TME including C-X-C motif ligand (CXCL)1, CXCL8, CXCL12, C-C motif ligand (CCL)1, CCL2, CCL3, CCL5, CCL7, and their corresponding receptors on MDSCs (C-C chemokine receptor [CCR]2, CCR5, and C-X-C chemokine receptor [CXCR]4 differently regulate the recruitment of MDSCs [11]. These chemokines are not specific to particular types of cancer and have a high degree of redundancy.

## 5. The Clinical Significance of MDSC in Ovarian Cancer Patients

### 5.1. The Frequency of MDSCs as a Prognostic Indicator or a Biomarker of Tumor Progression in Ovarian Cancer Patients

Increased numbers of circulating MDSCs have been detected in patients with various types of cancers [11,14,15]. Like in other cancer patients, as shown (Table 1), MDSCs were significantly increased in the peripheral blood mononuclear cells (PBMC), tumor or ascites in ovarian cancer patients [17,22,23,24,25,26,27,28,29,30,31,32,33]. Obermajer et al. and Cui et al. are the first to demonstrate an increased MDSC in the ascites [33] and ovarian tumors [32] of patients with ovarian cancer, respectively. Cui et al. also found that an increased number of tumor-infiltrating MDSCs was significantly associated with a short survival in patients with high-grade serous ovarian cancer [32]. Since then, an increasing number of reports have consistently suggested a strong association between increased MDSCs and decreased survival in ovarian cancer patients. Although the increments of both PMN- and M-MDSC have been observed in ovarian cancer patients (Table 1), some reports have suggested that M-MDSC might be a more reliable predictor of the clinical stage or survival in ovarian cancer patients, compared to PMN-MDSC [26,27,31,33].

Interestingly, two recent reports have suggested an association between decreased MDSCs and favorable treatment outcomes in ovarian cancer patients. Lee et al. showed that germline BRCA1 and 2 mutation-associated ovarian cancer, which is believed to have higher response rates to platinum-based chemotherapy than BRCA wild-type [34], has fewer circulating MDSCs and higher CD8^+^ T cells in PBMC compared with BRCA wild-type ovarian cancer [24]. Second, Li et al. demonstrated that metformin treatment in diabetic patients with ovarian cancer was associated with reduced circulating MDSCs, a concomitant increase in the circulating CD8^+^ T cells, and longer survival [29].

In an effort to investigate the cause of increased MDSC production in ovarian cancer, some groups have found that increased MDSCs were associated with increased levels of IL-6 and IL-10 in ascites [31], and leukocytosis [17]. Interestingly, a recent investigation conducted by Komura et al. has suggested that increased MDSC is observed in ovarian cancer patients with leukocytosis or in those whose ovarian tumors exhibit increased G-CSF expression [17]. The G-CSF has been widely used clinically during the course of chemotherapy to reduce the risk of chemotherapy-induced neutropenia. Although there have been no clinical reports suggesting that the exogenous G-CSF treatment have a negative impact on the survival of cancer patients, the impact of G-CSF treatment on the survival or the progression of patients with ovarian cancer has been undetermined. Thus, future studies are needed to evaluate the impact of exogenous G-CSF treatment on the induction of MDSC and survival in ovarian cancer patients receiving chemotherapy.

Although all of these studies have suggested the prognostic significance of MDSC in ovarian cancer, there have been many limitations in these studies such as small sample sizes, the inconsistent the use of inconsistent MDSC surface markers, or limited clinical information (patient characteristics, clinical stage, response to treatment, or survival rates). Moreover, the association between the histological subtypes of ovarian cancer and the number of MDSC has never been investigated. To employ MDSC as a prognostic indicator in the clinical management of ovarian cancer, further investigations will be required using a comprehensive epidemiological model.

### 5.2. In Vitro and In Vivo Investigation of MDSC in Ovarian Cancer

As in murine models of ovarian cancer (Table 2), MDSC production could be stimulated by various tumor-derived factors: CXCL1/2 [28], G-CSF [17], GM-CSF [35], VEGF [30], and PGE2 [33]. Moreover, MDSC have been shown to be recruited into the ovarian cancer TME through the CXCL1/2–CXCR2 [28] or CXCL12–CXCR4 axis [36].

In addition to its suppressive activity against CD8^+^ T cells in ovarian cancer TME [17,32,37], previous studies have demonstrated that MDSC increase the stem cell-like properties of ovarian cancer cells via by producing PGE2 [17], inducing the microRNA101 [32] or CSF2/STAT3 pathway activation [23]. Moreover, by producing PGE2, MDSCs increase the PD-L1 expression in ovarian cancer cells by activating AKT/mTOR signaling, which may facilitate ovarian cancer cells to escape destruction by the immune system [17].

## 6. Targeting MDSCs in Ovarian Cancer

### 6.1. Preclinical Investigation of MDSC-Targeting Therapies

In murine ovarian cancer models, as shown in Table 2, various strategies aiming at eliminating MDSCs from the TME have been evaluated: anti-Gr-1 antibody [17,30], anti-GM-CSF antibody [35], CXCR2 or CXCR4 antagonists [28,36], PGE2 or COX-2 inhibition [33], metformin [29], thrombin inhibitor [38], or bis-benzylidine piperidone RA190 [39]. They showed significant antitumor activities when used as monotherapies or in combination with chemotherapy. Moreover, recent efforts have demonstrated that MDSC-inhibition therapies targeting IL-10 or CXCR4 enhanced the therapeutic efficacy of anti-PD-1 therapy, leading to prolonged survival [37,40]. However, as the association between the histological subtypes of ovarian cancer and the number of MDSC remains undetermined, we cannot tell which histological subtypes are the potential candidates for MDSC-inhibition therapy in ovarian cancer patients.

### 6.2. Strategy to Inhibit Human MDSCs

In murine studies, the anti-Gr-1 antibody has been widely used to eliminate MDSCs from TME. However, due to the absence of a Gr-1 homologue in humans, anti-Gr-1 antibodies cannot be used in the clinical setting, and no specific inhibitors of human MDSC are currently available. At present, as shown in Table 3, various strategies to target MDSC have been proposed: (1) depletion of MDSC; (2) inhibition of MDSC functions; (3) prevention of MDSC recruitment into TME; and (4) promotion of the differentiation of MDSC into mature, nonsuppressive cells. Some of these MDSC-targeting strategies have already been tested in solid cancer patients and demonstrated significant activity to reduce the number of circulating or tumor-infiltrating MDSCs [42,43,44,45,46,47,48,49,50,51,52,53,54,55,56,57,58,59,60,61,62,63,64,65,66,67,68,69,70,71].

### 6.3. Clinical Trials Targeting MDSCs in Cancer Patients

Presently, biomarkers used to identify patients who might benefit from MDSC-targeting therapy have not been developed. However, as shown in Table 4, various clinical studies on MDSC-targeting therapies are currently underway either as a monotherapy, in combination with chemotherapy or as immune checkpoint inhibitors in patients with solid malignancies [72]. Hopefully, the next couple of years will bring exciting positive clinical data regarding MDSC-targeting therapies.

## 7. Conclusions

MDSCs are increased in ovarian cancer patients and play integral roles in disease progression. In order to inhibit their tumor-promoting effects, the efficacy of MDSC-targeting therapies (either as monotherapies or in combination with conventional treatments, including chemotherapy, radiotherapy, or anticancer immunotherapeutics) against ovarian cancer is currently being evaluated preclinically. We consider that increasing our understanding of MDSC biology will aid the development of optimal MDSC-targeting therapies, leading to the improvement of the prognosis of ovarian cancer patients.

## Figures and Tables

**Figure 1 cells-10-00329-f001:**
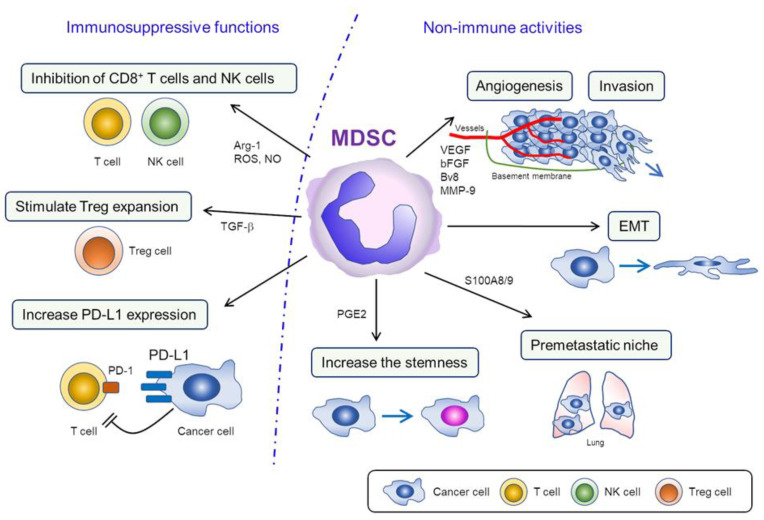
Functions of myeloid-derived suppressor cells (MDSC) in the tumor microenvironment (TME). 1) Immunosuppressive functions of MDSC; in TME, MDSC block T cell differentiation and inhibit the proliferation of T cell/NK cell by removing L-arginine. MDSC also inhibit the activities and diminish interferon response of T cell/NK cell via the production of ROS/NO. Increased PD-L1 expression in MDSC can lead to the downregulation of T cell function via the engagement of PD-1 in T cells. 2) Nonimmune activities of MDSC; MDSC promote cancer progression by inducing angiogenesis and tumor invasion via the production of VEGF, bFGF, Bv8, and MMP-9. Moreover, MDSC stimulate the metastatic activities of cancer cell by facilitating EMT and creating “premetastatic niches.” MDSC also induce “stemness” in certain cancers, which might be associated with resistance to chemotherapy or radiotherapy.

**Table 1 cells-10-00329-t001:** Summary of studies investigating the role of MDSC in ovarian cancer patients.

Author/Year	Histology	Samples Examined	Findings from Patient-Derived Samples
Komura et al. 2020 [17]	EOC	PBMC, Tumor	The proportion of MDSC in the peripheral blood or ovarian tumor was positively associated with the number of leukocytes and tumor G-CSF expression.
Li et al. 2020 [23]	Benign ovarian tumor EOC	PBMC	PMN-MDSC and M-MDSC were increased in ovarian cancer patients.
Lee et al. 2019 [24]	Stage III/IV or recurrent HGSOC	PBMC	gBRCAm was associated with increased CD8^+^ T cells and decreased MDSC.
Coosemans et al. 2019 [25]	Benign/borderline ovarian tumor or invasive EOC	PBMC	Increased MDSC was found to be an independent predictor of malignant disease.
Okta et al. 2019 [26]	Healthy donner EOC	PBMC, Tumor, Ascites	PMN-MDSC and M-MDSC were increased in ovarian cancer patients. Increased tumor-infiltrating M-MDSC was associated with advanced stage and decreased survival.
Santegoets et al. 2018 [27]	Healthy donner EOC	PBMC	M-MDSC was increased in ovarian cancer patients. Increased M-MDSC was associated with decreased survival.
Taki et al. 2018 [28]	EOC	Tumor	Increased tumor-infiltrating MDSC was associated with the high Snail expression.
Li et al. 2018 [29]	Healthy donner EOC	PBMC	Metformin treatment correlated with longer survival in diabetic patients with ovarian cancer, which was accompanied by a reduction in the circulating MDSC and a concomitant increase in the CD8^+^ T cells.
Rodrıguez-Ubreva et al. 2017 [22]	EOC	PBMC, Ascites	MDSC from patients displayed the MDSC-specific hypermethylation signatures.
Horikawa et al. 2017 [30]	HGSOC	Ascites	Increased MDSC was significantly associated with decreased intratumoral CD8^+^ T-cell infiltration and shorter survival.
Wu et al. 2017 [31]	Healthy donner EOC	PBMC, Ascites	M-MDSC was increased in the blood and ascites of ovarian cancer patients. Increased M-MDSC was associated with advanced stage, decreased survival, and increased serum IL-6/IL-10 concentration.
Cui et al. 2013 [32]	EOC	Tumor	Increased tumor-infiltrating MDSC was significantly associated with shorter survival.
Obermajer et al. 2011. [33]	EOC	Ascites	M-MDSC was increased in the ascites of ovarian cancer patients.

MDSC, myeloid-derived suppressor cells; EOC, epithelial ovarian cancer; PBMC, peripheral blood mononuclear cells; G-CSF, granulocyte-colony stimulating factor; PMN-MDSC, polymorphonuclear MDSC; M-MDSC, monocytic MDSC; HGSOC, high-grade serous ovarian cancer; gBRCAm, germline BRCA-mutation; CD8, cluster of differentiation 8; IL-6, interleukin 6; IL-10, interleukin 10.

**Table 2 cells-10-00329-t002:** Summary of in vitro/in vivo investigations of MDSC in ovarian cancer.

Author/Year	Experimental Models Employed		Findings from in vitro/in vivo Studies
Komura et al. 2020 [17]	Cell line	A2780, HM-1 cells	MDSC inhibited the activity of CD8^+^ T cells. PGE2 produced by tumor-derived G-CSF-induced MDSC increased the stem cell-like properties and tumor PD-L1 expression in ovarian cancer. Anti-Gr-1 antibody decreased serum PGE2 levels, MDSC in tumor, and cancer stem cells.
	Mouse type	BALB/c nude mice, B6C3F1	
	MDSC	Mouse and Patient-derived MDSC	
Li et al. 2020 [23]	Cell line	ES-2, SKOV3 and HO-8910	MDSC enhanced the stemness by activating CSF2/STAT3 pathway
	Mouse type	Not used	
	MDSC	Patient-derived MDSC	
Horikawa et al. 2020 [35]	Cell line	HM-1, ID-8 cells	Anti-VEGF therapy induced tumor hypoxia and GM-CSF expression, which recruited MDSC and inhibited tumor immunity. Anti-GM-CSF therapy reduced MDSC and improved the efficacy of anti-VEGF therapy.
	Mouse type	C57BL/6	
	MDSC	Mouse MDSC	
Zeng et al. 2019 [36]	Cell line	ID-8 cells	Inhibition of CXCL12-CXCR4 by a CXCR4 antagonist decreased tumor-infiltrating MDSC. Dual blockade of CXCL12-CXCR4 and PD-1-PD-L1 pathways prolonged survival.
	Mouse type	C57BL/6J	
	MDSC	Mouse MDSC	
Baert et al. 2019 [37]	Cell line	I-D8 cells	MDSC inhibited the activity of CD8^+^ T cells. Depletion of MDSC by anti-Gr-1 antibody improved survival.
	Mouse type	C57BL/6	
	MDSC	Mouse MDSC	
Taki et al. 2018 [28]	Cell line	HM-1, OVCAR8, OVCA433, A1847, JHOS2	Snail induced cancer progression via upregulation of CXCR2 and recruitment of MDSC. CXCR2 antagonist inhibited MDSC infiltration and delayed tumor growth.
	Mouse type	B6C3F1	
	MDSC	Mouse and Patient-derived MDSC	
Li et al. 2018 [29]	Cell line	Patient-derived ovarian cancer	Metformin blocked the suppressive function of MDSC by downregulating the expression and ectoenzymatic activity of CD39 and CD73 on MDSC.
	Mouse type	BALB/c-nu	
	MDSC	Mouse and Patient-derived MDSC	
Horikawa et al. 2017 [30]	Cell line	HM-1, ID-8 cells	MDSC migration and differentiation were augmented by VEGF signaling. Anti-Gr-1 antibody delayed the growth of tumor.
	Mouse type	C57BL/6 mice	
	MDSC	Mouse and Patient-derived	
Wu et al. 2017 [31]	Cell line	Not used	Ascites-derived IL-6 and IL-10 synergistically expanded M-MDSC in ovarian cancer.
	Mouse type	Not used	
	MDSC	Patient-derived MDSC	
Rodrıguez-Ubreva et al. 2017 [22]	Cell line	Not used	MDSC-Specific hypermethylation signature was mediated by PGE2-dependent DNMT3A overexpression in tumor.
	Mouse type	Not used	
	MDSC	Patient-derived MDSC	
Alexander et al. 2016 [38]	Cell line	ID-8	Dabigatran, a direct thrombin inhibitor, in combination with cisplatin decreased MDSC. Dabigatran augmented the antitumor activity of cisplatin.
	Mouse type	C57/Bl6	
	MDSC	Mouse MDSC	
Soong et al. 2016 [39]	Cell line	ID-8	Bis-benzylidine piperidone RA190 inhibited the activity of MDSC via the inhibition of STAT3 expression. MDSC-inhibition by RA190 delayed tumor growth, and prolonged survival.
	Mouse type	C57/Bl6	
	MDSC	Mouse MDSC	
Lamichhane et al. 2017 [40]	Cell line	ID-8	Combination of PD-1 blockade and IL-10 neutralization decreased tumor-infiltrating MDSC, delayed tumor growth, and prolonged survival.
	Mouse type	C57BL/6J	
	MDSC	Mouse MDSC	
Cui et al. 2013 [32]	Cell line	Patient-derived ovarian cancer	MDSC inhibited the activity of CD8^+^ T cells. MDSC enhanced stemness of cancer cells by inducing microRNA101.
	Mouse type	NSG (NOD/Scid/IL2Rγ)	
	MDSC	Patient-derived MDSC	
Obermajer et al. 2011 [33]	Cell line	Not used	PGE2 attracted MDSC into TME through the CXCL12-CXCR4 axis. PGE2 or COX2 inhibition blocked CXCL12 production and attenuated its ability to attract MDSC.
	Mouse type	Not used	
	MDSC	Patient-derived MDSC	

MDSC, myeloid-derived suppressor cells; CD, cluster of differentiation; PGE2, prostaglandin E2; G-CSF, granulocyte-colony stimulating factor; PD-L1, programmed death ligand 1; Gr-1, glutathione reductase 1; CSF2, colony stimulating factor 2; STAT3, signal transducer and activator of transcription 3; VEGF, vascular endothelial growth factor; GM-CSF, granulocyte macrophage colony-stimulating factor; CXCL, chemokine (C-X-C motif) ligand; CXCR, chemokine (C-X-C motif) receptor; PD-1, programmed death 1; M-MDSC, monocytic MDSC; PMN-MDSC, polymorphonuclear MDSC; IL, interleukin; DNMT3A, DNA (cytosine-5)-methyltransferase 3A; TME, tumor microenvironment; COX2, cyclooxygenase 2. Notes: All cell lines used, except for HM-1 and ID-8, are human ovarian cancer cells. HM-1 cells are mouse lymphnode-metastatic ovarian cancer, but the histological subtype has not been defined. ID-8 cells are mouse ovarian cancer cells histologically resembling human serous ovarian cancer, but a previous genomic analysis has shown that ID-8 is not an appropriate representative of human high-grade serous ovarian cancer [41].

**Table 3 cells-10-00329-t003:** Strategies for MDSC-targeting.

Treatment Strategies		Comments
(1) Depletion of MDSC.		
Induction of MDSC apoptosis	Chemotherapeutic agents	Gemcitabine [42], 5-FU [43], paclitaxel [44], cisplatin [45], docetaxel [46], and lurbinectedin [47]
	Tyrosine kinase inhibitors	Sunitinib [48] and sorafenib [49]
Inhibition of MDSC generation	IL-6 inhibitors	Anti-IL-6R mAb [50]
	CSF1R antagonists	GW2580 [51] and PLX3397 [52]
	S100A9 inhibitors	Tasquinimod [53]
	Diabetes drugs	Metformin [29]
	Thrombin inhibitor	Dabigatran [39]
(2) Inhibition of MDSC functions.		
	B-Raf inhibitor	Vemurafenib [54]
	Bisphosphonates	Zoledronic acid [55]
	PDE-5 inhibitors	Sildenafil, tadalafil, and vardenafil [56]
	STAT3 inhibitors	Stattic [19], CPA7 [57], S3I-201 [58], and AG490 [59]
	mTOR inhibitors	Rapamycin [60]
	PI3K inhibitors	IPI-145 [61] and IPI-549 [62]
	COX2 inhibitors	Celecoxib [63]
	NSAID	Nitroaspirin [64]
	HDAC inhibitor	Entinostat [65]
	IDO inhibitor	Indoximod [66]
(3) Prevention of MDSC recruitment into TME.		
	Chemokine receptor antagonists	AZD5069 (CXCR2) [67], Reparixin (CXCR2) [67], SX-682 (CXCR2) [67], AMD3100 (CXCR4) [67], CCX872 (CCR2) [68], and Maraviroc (CCR5) [67]
(4) Promoting the differentiation of MDSC into mature, nonsuppressive cells.		
	Vitamin A	ATRA [69]
	Vitamin D	1,25(OH)_2_D_3_ [70]
	Casein kinase inhibitor	Tetrabromocinnamic acid [71]
	Chemotherapeutic agents	Paclitaxel [44] and docetaxel [46]

MDSC, myeloid-derived suppressor cells; 5-FU, fluorouracil; IL-6, interleukin 6; IL-6R, interleukin 6 receptor; CSF1R, colony stimulating factor 1 receptor; S100A9, S100 calcium-binding protein A9; PDE-5, phosphodiesterase 5; STAT3, signal transducer and activator of transcription 3; mTOR, mammalian target of rapamycin; PI3K, phosphoinositide 3-kinase; COX2, cyclooxygenase 2; NSAID, nonsteroidal anti-inflammatory drug; HDAC, histone deacetylase; IDO, indoleamine 2,3-dioxygenase; TME, tumor microenvironment; CXCR, chemokine (C-X-C motif) receptor; CCR, chemokine (C-C motif) receptor; ATRA, all-trans retinoic acid.

**Table 4 cells-10-00329-t004:** Clinical trials targeting MDSC in cancer patients. (available from ClinicalTrials.gov).

Trial Number *	Purpose/Design of the Study	Conditions	Interventions
NCT04022616	Examine MDSC frequency	Breast cancer	Specimen collection procedure
NCT02868255	Examine MDSC frequency	Hepatocellular carcinoma Ovarian cancer	Specimen collection procedure
NCT02735512	Examine MDSC frequency	Bladder cancer	Specimen collection procedure
NCT02664883	Examine MDSC frequency	Renal cell cancer	Specimen collection procedure
NCT04387682	Examine MDSC frequency	Oral squamous cell carcinoma	Specimen collection procedure
NCT02669173	A phase I study investigating that suppression of MDSCs with low dose capecitabine is safe and feasible.	Glioblastoma	Capecitabine plus bevacizumab
NCT01803152	A phase I study consisting with 2 parts: after the dose escalation study of dendritic cell (DC) vaccination, the safety, feasibility and the effect of MDSC inhibition using gemcitabine concurrently with DC vaccination will be evaluated.	Sarcoma	DC vaccine, gemcitabine, imiquimod
NCT03525925	A phase I trial evaluating the safety of ibrutinib and nivolumab combination therapy and determine the effect of ibrutinib on circulating levels of MDSC.	Metastatic malignant solid neoplasm	Ibrutinib plus nivolumab
NCT02637531	A phase I study evaluating the safety and the tolerability of IPI-549 in combination with nivolumab.	Advanced solid tumors with increased MDSC	IPI-549 plus nivolumab
NCT03161431	A phase I study evaluating the optimal dose of SX-682 with or without pembrolizumab, and the inhibitory effect of SX-682 on MDSC.	Melanoma	SX-682 and pembrolizumab

* Available from ClinicalTrials.gov; https://www.clinicaltrials.gov/ (accessed on 21 November 2020). MDSC, myeloid-derived suppressor cells; DC, dendritic cell.

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
