# Peer review of "Targeting Myeloid-Derived Suppressor Cells in Ovarian Cancer"

_cells, 2021, doi:10.3390/cells10020329_

Round 1

Reviewer 1 Report

This article comprehensively reviews current evidence, clinical roles, potential targeting therapies on MDSC. Although it is well-written, there are a few minor points to be revised before publication.

Minor comments; 

Line 99, Figure 1 legend; Does MDSC not induce but inhibit T cell differentiation, as authors described in Line 82-85?

Line 183 and Table 1; Regarding prognostic significance of MDSC, it would be worth to describe the differences between ovarian cancer histotypes in detail. This might explain which histotype is potential candidates for MDSC targeting strategies.

Line 188-199, Section 5.2; Please describe the detail of murine models of ovarian cancer on references listed in Table 2. If the references used ID-8 as ovarian cancer model, authors should add the fact that genomic analysis has shown that ID8 tumors are not appropriate representative of human HGSOC (Cancer Res. 2016 Oct 15;76(20):6118-6129. doi: 10.1158/0008-5472.CAN-16-1272. PMID: 27530326).

Line 217-227, Section 6.2, Table 3; As you described here, chemotherapies induce MDSC apoptosis, and G-SCF stimulates G-CSF in vitro and in vivo model in Section 5.2. Please add current evidence about the effect of G-CSF support during chemotherapy on MDSC induction and poor prognosis.

Line 234-240; Please write down each potential MDSC targeting therapy in manuscript concretely; listed in Table 4, from NCT02669173 to NCT03161431.

Author Response

Responses to reviewers #1:

Comment 1: Line 99, Figure 1 legend; Does MDSC not induce but inhibit T cell differentiation, as authors described in Line 82-85?

Response: We have corrected an error (line 101).

Comment 2: Line 183 and Table 1; Regarding prognostic significance of MDSC, it would be worth to describe the differences between ovarian cancer histotypes in detail. This might explain which histotype is potential candidates for MDSC targeting strategies.

Response: Thank you for the reviewer’s thoughtful comment. We have read through the papers listed in Table 1. However, although various epithelial ovarian cancers were included in the investigations, no researches examined the association between histological subtypes and MDSC number. Thus, at this point, we cannot tell which histological subtypes are potential candidates for MDSC-inhibition therapy. We have indicated these in lines 198-200 and 224-227 of the revised manuscript.

Comment 3: Line 188-199, Section 5.2; Please describe the detail of murine models of ovarian cancer on references listed in Table 2. If the references used ID-8 as ovarian cancer model, authors should add the fact that genomic analysis has shown that ID8 tumors are not appropriate representative of human HGSOC (Cancer Res. 2016 Oct 15;76(20):6118-6129. doi: 10.1158/0008-5472.CAN-16-1272. PMID: 27530326).

Response: As suggested, we have described the detail of murine models of ovarian cancer on references listed in Table 2. As ID-8 had been employed as ovarian cancer model in various investigations, we indicated the fact that genomic analysis has shown that ID-8 tumors are not appropriate representative of human HGSOC (lines 234-236).

Comment 4: Line 217-227, Section 6.2, Table 3; As you described here, chemotherapies induce MDSC apoptosis, and G-SCF stimulates G-CSF in vitro and in vivo model in Section 5.2. Please add current evidence about the effect of G-CSF support during chemotherapy on MDSC induction and poor prognosis.

Response: As suggested, we have included a current evidence about the effect of G-CSF support during chemotherapy on MDSC induction and poor prognosis (lines 186-194).

Comment 5: Line 234-240; Please write down each potential MDSC targeting therapy in manuscript concretely; listed in Table 4, from NCT02669173 to NCT03161431.

Response: As suggested, we have concretely described each potential MDSC targeting therapy listed in Table 4, from NCT02669173 to NCT03161431.

Responses to reviewers #2:

Comment 1: It would be more informative if authors can provide more clear rationale as to why MDCS is a good therapeutic target for ovarian cancer.

Response: As suggested, we more clearly stated a rationale for the MDCS-targeting in the treatment of ovarian cancer (lines 52-55).

Comment 2: In section 1 MDSC development and activation, more detailed description is needed on how MDSC maturation by factors secreted from the tumor tissues. Currently, factors are merely listed and their biological significance or mechanisms are not described or described in an insufficient manner.

Response: Thank you for the reviewer’s thoughtful comment. We have included a more detailed description on the effect of tumor-derived factors on the differentiation of MDSC (lines 129-145).

Comment 3: On page 4 line 147 in section 2 Recruitment of MDSC into TME, a more detailed explanation or references of what the authors refer to as "unique distribution of chemokines in TME" are needed.

Response: Although we mentioned “the unique distribution of chemokines in the TME may help maintain a constant supply of different MDSC subset (M- or PMN-) into TME”. The “unique distribution of chemokines in the TME of specific cancer” has not been demonstrated. Moreover, the association between the distribution of specific chemokines and different MDSC subset (M- or PMN-) has also been unknown. Accordingly, this sentence is my personal view. To avoid confusion, we deleted this sentence in the revied version.

Comment 4: In Table 3 is currently unorganized and bit clustered, thus the information in the table should be categorized into 4 parts as described in the main section of the manuscript (depletion of MDSC, inhibition of MDSC functions, prevention of MDSC recruitment into TME, and promotion of the differentiation of MDSC into mature cells).

Response: As suggested, we have revised Table 3.

Reviewer 2 Report

Title: Targeting myeloid-derived suppressor cells in ovarian cancer

Seiji Mabuchi

Department of Obstetrics and Gynecology, Kaizuka City Hospital, Osaka, Japan

[email protected]

General comments:

Mabuchi et al. have reviewed the background of MDSC, its clinical significance, and the treatment strategy for targeting MDSC in ovarian cancer. The manuscript is well-organized but the contents are too compact. Therefore, some issues need to be supplemented prior to publication in the Cells.

Specific comments:

  1. It would be more informative if authors can provide more clear rationale as to why MDCS is a good therapeutic target for ovarian cancer.
  2. In section 1 MDSC development and activation, more detailed description is needed on how MDSC maturation by factors secreted from the tumor tissues. Currently, factors are merely listed and their biological significance or mechanisms are not described or described in an insufficient manner.
  3. On page 4 line 147 in section 2 Recruitment of MDSC into TME, a more detailed explanation or references of what the authors refer to as "unique distribution of chemokines in TME" are needed.
  4. In Table 3 is currently unorganized and bit clustered, thus the information in the table should be categorized into 4 parts as described in the main section of the manuscript (depletion of MDSC, inhibition of MDSC functions, prevention of MDSC recruitment into TME, and promotion of the differentiation of MDSC into mature cells).

Author Response

(The authors gave the same response as above.)
